# Functional Heterogeneity and Therapeutic Targeting of Tissue-Resident Memory T Cells

**DOI:** 10.3390/cells10010164

**Published:** 2021-01-15

**Authors:** Esmé T. I. van der Gracht, Felix M. Behr, Ramon Arens

**Affiliations:** Department of Immunology, Leiden University Medical Center, 2333 ZA Leiden, The Netherlands; E.T.I.van_der_Gracht@lumc.nl (E.T.I.v.d.G.); F.M.Behr@lumc.nl (F.M.B.)

**Keywords:** T cells, heterogeneity, tissue residency, immunotherapy, therapeutic targeting

## Abstract

Tissue-resident memory T (T_RM_) cells mediate potent local innate and adaptive immune responses and provide long-lasting protective immunity. T_RM_ cells localize to many different tissues, including barrier tissues, and play a crucial role in protection against infectious and malignant disease. The formation and maintenance of T_RM_ cells are influenced by numerous factors, including inflammation, antigen triggering, and tissue-specific cues. Emerging evidence suggests that these signals also contribute to heterogeneity within the T_RM_ cell compartment. Here, we review the phenotypic and functional heterogeneity of CD8^+^ T_RM_ cells at different tissue sites and the molecular determinants defining CD8^+^ T_RM_ cell subsets. We further discuss the possibilities of targeting the unique cell surface molecules, cytokine and chemokine receptors, transcription factors, and metabolic features of T_RM_ cells for therapeutic purposes. Their crucial role in immune protection and their location at the frontlines of the immune defense make T_RM_ cells attractive therapeutic targets. A better understanding of the possibilities to selectively modulate T_RM_ cell populations may thus improve vaccination and immunotherapeutic strategies employing these potent immune cells.

## 1. Introduction

CD8^+^ T cells play a crucial role in immune protection against pathogens and cancer [1]. Upon antigen recognition following vaccination or infection, naïve antigen-specific CD8^+^ T cells clonally expand and differentiate into effector cell populations. After an initial proliferative expansion, effector T cells eventually differentiate into phenotypically diverse memory CD8^+^ T cell subsets. Central memory T (T_CM_) cells are characterized by expression of the lymph node homing molecules CD62L and CCR7, which enable them to patrol secondary lymphoid organs, whereas effector memory T (T_EM_) cells lack these homing molecules and circulate throughout the body [2]. In addition to the circulating T_CM_ and T_EM_ cells, tissue-resident memory CD8^+^ T (T_RM_) cells form an integral part of the memory T cell pool. In contrast to their circulating counterparts, T_RM_ cell populations localize to peripheral tissues and lymphoid organs, where they reside with minimal egress [3,4,5]. T_RM_ cells are characterized by the expression of tissue retention molecules, such as CD69, and downregulation of genes involved in tissue egress, including *S1pr1* and *Ccr7* [6]. Upon restimulation, T_RM_ cells rapidly release effector molecules, such as interferon-γ (IFN-γ), tumor necrosis factor (TNF), and granzyme B; initiate proliferation; and induce recruitment of other immune cells to the site of challenge [7,8]. T_RM_ cells are found in a large variety of tissues and mediate potent local innate and adaptive immunity against pathogens and tumors [9,10].

Defining the crucial molecular interactions and mechanisms that induce effective CD8^+^ T cell memory pools at the right location is critical for the design of successful vaccines against cancer and infectious disease. Here, we review the heterogeneous phenotypes and functions of CD8^+^ T_RM_ cells and how their unique characteristics, including cell surfaces molecules, cytokine and chemokine receptors, transcription factors, and metabolic features, can be targeted to improve immunotherapies.

## 2. T_RM_ Cell Formation and Maintenance

### 2.1. T_RM_ Cell Formation

Both CD8^+^ T_RM_ cells and circulating memory T cells appear to arise primarily from the KLRG1^lo^ IL-7Rα^hi^ memory precursor effector cell pool [6,11]. However, it remains incompletely understood at which stage the tissue-resident and circulating lineages separate. Transcriptional analysis revealed that the gene expression profile of T_RM_ cells is already largely established in effector CD8^+^ T cells at peripheral tissue sites during the effector phase after infection, indicating a role of the tissue microenvironment in instructing T_RM_ cell formation [11]. The development of T_RM_ cells is shaped by the cytokine milieu, with the cytokines IL-12, type I IFN, and IL-15 playing important roles in the differentiation of these memory T cells [12,13,14]. In addition, various transcription factors, including Hobit and its homolog Blimp-1 as well as Runx3, Notch, and Bhlhe40, regulate CD8^+^ T_RM_ cell development [11,15,16,17]. Blimp-1 and Hobit mediate tissue retention of CD8^+^ T_RM_ cells by suppressing tissue egress pathways. These transcription factors regulate the tissue residency of CD8^+^ T_RM_ cells and other tissue-resident lymphocyte populations, including liver-resident natural killer T (NKT) cells and type 1 innate lymphoid cells (ILC1s) [17]. Transcriptional regulation of T_RM_ cells is further characterized by tissue-specific adaptations [6,17]. CD8^+^ T_RM_ cell formation in the lungs after influenza virus infection primarily depends on Blimp-1 rather than Hobit, potentially through control of the lineage choice between T_CM_ and T_RM_ cells during the differentiation of virus-specific CD8^+^ T cells [18].

### 2.2. T_RM_ cell Maintenance and Plasticity

In mice, CD8^+^ T_RM_ cells persist in many tissues for extended periods of time, including the skin, liver, and intestine [5,13,19]. Experimental procedures such as parabiosis, where the blood circulation of two animals is conjoined, and transplantation experiments have demonstrated the minimal recirculation capacity of these cells under steady-state conditions [4,5,20]. The role of T_RM_ cells has been studied in experimental settings involving selective depletion of these cells (e.g., by using CXCR3 antibody or NAD treatment), sequestration of circulating cells in the lymph nodes by FTY720 treatment, and comparison of settings in which T_RM_ cells were present and/or absent in tissues prior to (local) challenges [19,21,22]. While research on T_RM_ cells in human tissues is more challenging, studies of human tissue transplants demonstrated that T_RM_ cells also persist in human tissues for long periods of time [23,24].

Even though T_RM_ cells persist for years in many tissues, the population dynamics of T_RM_ cells in specific tissues remain incompletely understood. Maintenance of T_RM_ cells is influenced by the tissue microenvironment, including homeostatic cytokines such as IL-7, IL-15, and transforming growth factor-β (TGF-β). In addition, maintenance of CD8^+^ T_RM_ cells is regulated by tissue damage via P2RX7, a damage/danger-associated molecular pattern receptor that is triggered by extracellular nucleotides (ATP, NAD^+^). P2RX7 activation in vivo by exogenous NAD^+^ or tissue damage leads to specific depletion of T_RM_ cells while retaining circulating T cells [25]. The tissue microenvironment also influences the metabolic requirements of CD8^+^ T_RM_ cells, which shape their survival and functionality. Specifically, skin T_RM_ cells require the uptake and metabolism of exogenous free fatty acids (FFA) to persist in the tissue [26]. Following viral infection, developing CD8^+^ T_RM_ cells in the skin differentially express the fatty-acid-binding proteins 4 and 5 (FABP4 and FABP5), which mediate lipid uptake and intracellular transport. Recently, Frizzell and colleagues found that T_RM_ cells from different tissues show varying patterns of isoform usage of fatty-acid-binding proteins, which is determined by tissue-derived factors [27]. Together, these findings indicate that T_RM_ cells undergo metabolic adaptations shaped by the local microenvironment.

Local CD8^+^ T_RM_ cell populations are further characterized by quantitative and qualitative changes following local antigen encounter. In the skin and female reproductive tract, CD8^+^ T_RM_ cells undergo local proliferation in situ within days after antigen encounter [28,29]. Here, preexisting T_RM_ cell populations within the peripheral tissues appear to be the primary source of both local proliferative recall responses and secondary T_RM_ cells developing after reinfection [28,29,30]. Additionally, T_RM_ cell reactivation triggers the recruitment of circulating memory CD8^+^ T cells into the tissue, which undergo antigen-independent T_RM_ cell differentiation in situ. Circulating memory CD8^+^ T cells may differ in their capacity to form T_RM_ cells at different tissue sites. Predominantly, CD8^+^ T_EM_ cells are able to convert into liver CD8^+^ T_RM_ cells upon adoptive transfer in the presence of antigen [31]. In contrast, both T_EM_ and T_CM_ cells are limited in their capacity to form CD103^+^ T_RM_ cells at mucosal sites after reinfection [32].

More recently, a study using a unique T_RM_ fate-mapping model demonstrated that T_RM_ cells are able to reenter circulation upon reactivation [29]. Following pathogen rechallenge, T_RM_ cells downregulate Hobit expression on antigen encounter and form circulating memory T cells. These tissue-experienced ex-T_RM_ cells are transcriptionally and functionally distinct from secondary T_EM_ cells. Thus, subsets of T_RM_ cells appear to maintain recirculation capacity and thereby contribute to systemic immunity following reinfection. In line with these findings, Fonseca and colleagues found that local reactivation of T_RM_ cells precipitates egress to circulation [33]. Even though these ex-T_RM_ cells share developmental plasticity with other circulating memory subsets, they remain epigenetically poised for tissue migration and T_RM_ cell redifferentiation. Tissue-experienced ex-T_RM_ cells may therefore represent functional adaptions of the memory CD8^+^ T cell compartment to reinvading pathogens.

### 2.3. T_RM_ Cell Heterogeneity

Transcriptional profiling has revealed a unique gene expression signature of T_RM_ cells shared between T_RM_ cells at various locations [6,17]. However, T_RM_ cells in different organs are characterized by tissue-specific gene expression profiles [6,17]. Recently, two studies identified previously unknown transcriptional heterogeneity within the developing and established T_RM_ cell compartments of the intestine, representing functionally distinct T_RM_ subsets with divergent transcriptional programs, cytokine production, and secondary memory potential [34,35]. Recent evidence additionally suggests that memory CD8^+^ T cell heterogeneity of both circulating and T_RM_ cells is primarily influenced by pathogen-specific cues and further shaped by the tissue microenvironment [36]. However, the functional consequences of CD8^+^ T_RM_ cell heterogeneity remain incompletely understood.

Local inflammatory microenvironments regulate the differentiation and persistence of CD8^+^ T_RM_ cells. In the intestine, TGF-β and proinflammatory chemokines differentially control the formation of CD103^+^ and CD103^−^ T_RM_ cell subsets with distinct localization and functionality [13]. Intestinal macrophages produce the inflammatory cytokines IL-12 and IFN-β, which drive the persistence of CD103^-^ T_RM_ cells, while CD103^+^ T_RM_ cells develop independently [14]. In the lungs, a distinct subset of T_RM_ cells in the interstitium acts as a resident reservoir and maintains airway T_RM_ cells [37]. CD8^+^ T_RM_ cells in the lung airways are seeded continuously by T_RM_ cells from the lung interstitium, a process driven by the chemokine receptor CXCR6. In addition to the cytokine environment, the functional properties of T_RM_ cells can be influenced by cellular interactions. Low et al. found that numerous hematopoietic and non-hematopoietic antigen-presenting cells were able to reactivate lung T_RM_ cells but the quality of functional T_RM_ cell responses, as defined by activation signatures, depended on the identity of the antigen-presenting cells [38]. In the skin, CD49a distinguishes a subset of T_RM_ cells poised for IFN-γ production and cytotoxic capacity while CD49a^−^ skin T_RM_ cells preferentially produce IL-17. Importantly, this functional heterogeneity between CD49a^+^ and CD49a^−^ cells is preserved in inflammatory diseases such as vitiligo and psoriasis [39]. Thus, multiple factors, including the cytokine environment, and antigen-triggering and cellular interactions impact the heterogeneity of local CD8^+^ T_RM_ cell populations.

## 3. Functional Heterogeneity and Therapeutic Targeting

Circulating memory CD8^+^ T cells are heterogeneous in their phenotype and migratory properties as well as their functions and proliferative potential [40]. As reviewed by Jameson and Wherry, the nomenclature of memory T cells is challenging due to the continuum of memory T cell properties with respect to their location and recirculation capacity, metabolism, epigenetic regulation, and longevity [41]. Hence, in addition to the main circulating T_EM_ and T_CM_ subsets, many more subdivisions can be defined, including but not limited to stem cell memory T (T_SCM_) cells and exhausted T (T_EX_) cells.

Next to the heterogeneous population of circulating memory CD8^+^ T cells, resident CD8^+^ T cells also display phenotypic and functional heterogeneity. CD8^+^ T_RM_ cells can, for example, display activated phenotypes, as characterized by high amounts of markers related to costimulatory or inhibitory receptors, cytokine receptors, and/or chemokine receptors. Since T_RM_ cells mediate tissue immunity, enhancing their numbers or functional properties could be beneficial for viral immunity and immunotherapy in malignant diseases.

In many human cancers, large quantities of tumor-infiltrating T_RM_-like cells correlate with an improved overall survival [42,43,44]. In addition, T_RM_ cells play a crucial role in controlling tumor outgrowth in preclinical models [45,46]. Here, T_RM_ cells may amplify antitumor responses through dendritic cell activation [47]. Thus, the possibility to increase T_RM_ cell numbers and thereby enhance local protective immunity constitutes a promising immunotherapeutic approach. An increase in T_RM_ cell numbers may be achieved by providing antigen and/or inflammation at the desired tissue site. For instance, intranasal, but not systemic administration of live-attenuated influenza virus generates virus-specific CD8^+^ T_RM_ cells in the lungs that mediate heterosubtypic protection against influenza virus infection [48]. Interestingly, lung T_RM_-mediated protection against influenza virus was independent of circulating T cells and neutralizing antibodies and persisted long-term after vaccination. In preclinical models of melanoma, intradermal but not intraperitoneal vaccination generated skin T_RM_ cells mediating strong protection against cutaneous melanoma [46]. The localized formation of T_RM_ cells can thus provide immunity to both infectious and malignant challenges [49]. In addition, T_RM_ cell responses could be enhanced by active recruitment of T cells to the site of inflammation. Local inflammation can be sufficient to induce the development of protective T_RM_ cell populations at peripheral sites in the absence of a local antigen [50]. However, T_RM_ cell formation is substantially enhanced in the presence of cognate antigen within the tissue microenvironment [51,52,53]. Similarly, sequential immunizations result in global lodgment of skin T_RM_ cells, but active recruitment by local inflammation or infection to specific tissue sites enhances local skin T_RM_ cell numbers and provides maximal protection against pathogen challenge, demonstrating the potency of localized T_RM_ cell deposition as a means of pathogen control [53,54]. Another possibility to enhance the expansion of CD8^+^ T_RM_ cells is topical application of chemokines or antigenic peptides to locally boost cellular immunity [55]. Moreover, T_RM_ cell-mediated protection can be provoked upon adoptive cell therapy (ACT) through differentiation of the transferred cells into T_RM_ cells via pathogen-based vaccines or targeting transcription factors that promote tissue residency [11,56]. Hereafter, we focus on the phenotypic and functional features of CD8^+^ T_RM_ cell heterogeneity and how these cells can be targeted to improve immunity (summarized in Figure 1).

### 3.1. CD69 and CD38

CD69, a transmembrane C-type lectin protein, is transiently induced early after activation on T cells [57]. However, in contrast to the transient expression of this molecule on other T cell subsets, the vast majority of T_RM_ cells in most tissues constitutively express CD69 under resting conditions. Functionally, CD69 regulates peripheral T cell retention by inhibiting sphingosine-1-phosphate receptor 1 (S1PR1) expression and function [57,58]. T cells require S1PR1 to sense S1P gradients, which induces chemotactic migration of T cells. Inflammatory cytokines, including TGF-β, interleukin-33 (IL-33), and TNF, also mediate suppression of S1PR1 expression via downregulation of the transcription factor Krüppel-like factor 2 (KLF2) [59]. Modulating these inflammatory factors during T cell differentiation may thus constitute a way to suppress tissue egress of T cells and enhance T_RM_ cell formation.

CD69 can be directly targeted using specific antibodies, and intravenous administration of anti-CD69 antibodies increased leukocyte numbers in secondary lymphoid organs and enhanced anti-vaccinia virus immunity [60]. Here, targeting CD69 primarily increased the number of IFN-γ- and TNF-producing NK and circulating T cells, suggesting broad effects of CD69 targeting outside of the resident T cell compartment. Moreover, CD69 is also expressed on hematopoietic progenitor cells. Targeting of CD69 can induce mobilization and proliferation of hematopoietic stem and progenitor cells leading to bystander T cell proliferation through pDC-mediated IL-2 production [61,62]. Thus, the rapid and widespread induction of CD69 on immune cells upon infection and inflammation makes specific targeting of T_RM_ cells via CD69 challenging. It remains, however, to be determined whether CD69 targeting affects T_RM_ cell expansion and function during homeostatic conditions. CD38, a glycoprotein that functions as an enzyme and receptor, is also highly expressed by T_RM_ cells [63]. However, CD38 is expressed by many other immune cells as well. Consequently, the broad expression of CD69 and CD38 make these molecules less attractive for specific targeting of T_RM_ cells.

### 3.2. Integrins

Integrins, such as CD49a (α1β1) and CD103 (αEβ7), mediate tissue retention of CD8^+^ T_RM_ cells. These integrins are characterized by different functions and modes of action and may show differential expression on mouse and human CD8^+^ T_RM_ cells. While CD103 is absent on liver CD8^+^ T_RM_ cells in mice, it is expressed by human hepatic CD8^+^ T_RM_ cell subsets [64]. CD103 binds E-cadherin, which is commonly expressed by epithelial cells [65,66]. CD103 has been associated with cytotoxic CD8^+^ T cell responses in several human pathologies, including autoinflammatory diseases and cancer [42,67,68]. In line with this, CD103^+^ T_RM_-like cells in tumors are characterized by a cytotoxic signature and are able to secrete inflammatory cytokines [11,43]. High frequencies of tumor-infiltrating CD103^+^ T cells are associated with improved survival rates of patients in various solid cancers [42,43,44], and CD103 may directly contribute to improved immune control of these malignant diseases. The interaction of CD103 with E-cadherin on tumor cells enhances cytokine production by antitumor T cells, and targeting CD103 with blocking antibodies inhibits cytotoxicity of CD103^+^ T cells towards tumor cells [68,69]. In addition, CD103 expression by skin T_RM_ cells appears to be required for their protection against tumors [70]. Enhancing the frequency of CD103^+^ T_RM_ cells thus appears to constitute an attractive approach to combat infectious and malignant diseases. CD103 may be targeted directly via specific antibodies or by modulating TGF-β signaling, which induces CD103 expression on T cells. Using anti-CD103 and anti-CD49a blocking antibodies, knockout mice, and intravital imaging, Reilly and colleagues showed that CD103 and CD49a differentially support adherence and motility of virus-specific T cells after the resolution of an influenza virus infection [71]. Moreover, enhancing CD103 expression on mucosal T cells by targeting DC-dependent activation of TGF-β has been shown to inhibit tumor progression in a preclinical model of breast cancer [72].

CD49a, also known as very late antigen-1 (VLA-1), binds collagen within the extracellular matrix to establish the tissue residency of T_RM_ cells [73,74]. Recently, Bromley and colleagues showed that CD49a supports CD8^+^ T_RM_ persistence within the skin and increases the frequency of IFN-γ^+^ CD8^+^ T_RM_ cells following local antigen challenge [75]. This suggests that CD49a acts as a costimulatory receptor and/or regulator of CD8^+^ T_RM_ cell migration to increase antigen encounter in vivo. CD49a expression further defines CD8^+^ T_RM_ cells in human skin that are poised for cytotoxic function [39]. Here, CD49a^+^ T_RM_ cells produce IFN-γ, whereas CD49a^-^ T_RM_ cells produce IL-17. In addition, CD49a^+^ T_RM_ cells from healthy skin rapidly induce the expression of the effector molecules perforin and granzyme B when stimulated with IL-15, thereby promoting a strong cytotoxic response. Similar to CD103, CD49a expression on tumor-infiltrating T_RM_ cells correlates with improved survival of cancer patients [76]. Antibody-mediated blockade of CD49a in turn impairs tumor control in a preclinical model [76], highlighting the functional importance of CD49a in tumor immunity. The expression of CD49a and CD103 can be induced by different cytokines, which could provide opportunities to modulate the expression of these adhesion molecules and the corresponding functional subsets of T_RM_ cells by cytokine therapies [77]. In addition to CD49a and CD103, CD8^+^ T_RM_ cells can upregulate LFA-1, a heterodimer composed of two members of the integrin family: αL (CD11a) and β2 (CD18). Using intra-vital imaging, it was found that CD8^+^ T_RM_ cells patrolling in the hepatic sinusoids are dependent on LFA-1–ICAM-1 interactions [78].

### 3.3. Inhibitory Receptors

Following activation, both circulating and T_RM_ cells (transiently) express inhibitory receptors, such as PD-1, CTLA-4, and LAG-3 (CD223), which serve as immune checkpoints by dampening further T cell activation. The expression of these inhibitory molecules is associated with CD8^+^ T cell exhaustion, and consequently, many different immune-therapeutics target inhibitory immune checkpoints to improve cancer immunotherapy [79]. Evidently, during chronic viral infection, T_RM_ cells express PD-1 along with other exhaustion-associated molecules such as LAG-3 [80,81,82]. However, even in the absence of persistent antigen triggering, T_RM_ cells express a wide variety of inhibitory receptors [6,83].

In non-small cell lung carcinoma, CD103^+^CD8^+^ T cells co-express CD49a and CD69 and display a molecular profile characterized by the expression of PD-1 and the ectonucleotidase CD39 [84]. Genes involved in T cell exhaustion, including BTLA, LAG-3, and TIGIT, were more strongly expressed in CD8^+^ T_RM_ cells than in KLRG1^+^ tumor-infiltrating lymphocyte (TIL) subsets. The co-expression of CD39 together with CD103 has been reported to identify tumor-reactive CD8^+^ T cells in solid human tumors [85]. Here, prolonged T cell receptor (TCR)stimulation in the presence of TGF-β was necessary for the co-expression of CD103 and CD39 on CD8^+^ T cells. CD39^+^CD8^+^ T cells display divergent functional capacities and increased PD-1 expression, and are characterized by reduced expression of inflammatory cytokines (IFN-γ and IL-2) and cytotoxic mediators (perforin and granzyme B) [86]. In addition, CD39^+^ CD8^+^ TILs exhibited suppressive function in vitro. Modulation of CD39 could provide a promising strategy to restore the suppressive CD8^+^ T_RM_ cells in these settings, as compounds inhibiting CD39-related ATPases can improve CD39^+^ CD8^+^ T cell effector function ex vivo [86].

Besides circulating T cells, CD8^+^ T_RM_ cells also play an important role in the efficacy of PD-1/PD-L1 checkpoint blockade. Upon PD-1 blockade, CD103^+^ CD8^+^ T_RM_ cells accumulate in tumors of therapy-responding lung cancer patients, and these cells display enhanced proliferation and cytotoxicity toward cancer cells [84]. In non-small cell lung carcinoma, tumor-specific CD103^+^CD8^+^ TILs, expressing PD-1 and Tim-3, constitute a highly activated subpopulation and represent a prognostic factor for survival [42]. In addition to increasing the proliferation of responding T cells, the PD-L1 blockade also enhances IFN-γ and TNF production by antigen-specific CD8^+^ T_RM_ cells [87]. Moreover, the PD-L1 blockade induces selective expansion of tumor-infiltrating CD4^+^ and CD8^+^ T cell subsets. These subsets co-express both activating (ICOS) and inhibitory (LAG-3 and PD-1) molecules [88], and co-targeting of these molecules further improve the efficacy of the PD-1 blockade, thus demonstrating the synergistic effect of combined immunotherapies. Similarly, a combined blockade of PD-1 and LAG-3 can enhance the frequencies and functionality of virus-specific CD8^+^ T_RM_ cells, thereby reducing disease severity following herpes simplex virus infection [89].

Although the upregulation of PD-1 in cancer and chronic infection is associated with T cell exhaustion, the physiological role of PD-1 is not negative by definition. For example, T_RM_ cells mediate immune homeostasis in the human pancreas through the PD-1/PD-L1 pathway [90]. In addition, the expression of inhibitory receptors, including PD-1 and Tim-3, may prevent T_RM_-mediated immunopathology in chronic inflammatory diseases [91]. In the human brain, CD8^+^ T_RM_ cells expressing PD-1 provide protection against neurotropic virus reactivation [92]. Moreover, subsets of PD-1^+^ T cells possess a self-renewal capacity, and this particular population responds well to PD-1 blockade, indicating functional heterogeneity within the PD-1^+^ T cell pool.

Another cell surface molecule that can be expressed by CD8^+^ T_RM_ cells is CTLA-4 (CD152), a receptor that mediates inhibitory signals, by counteracting co-stimulation provided via CD28 by competing with CD28 for binding the shared ligands CD80/CD86 [93,94], or by actively removing these ligands via internalization for degradation [95]. Blocking CTLA-4 is thought to promote the priming of CD8^+^ T cells [96,97]. Targeting of CTLA-4 during priming indeed induces a circulating CD8^+^ T_EM_ phenotype but also results in increased CD8^+^ T_RM_ cells, leading to enhanced protection in subcutaneous tumor models [31].

The therapeutic effect of targeting inhibitory receptors may be further potentiated by cytokines. Tumor-derived IL-33 increases the number and function of CD103^+^CD8^+^ TILs, and a combination of IL-33 with the CTLA-4 and PD-1 immune checkpoint blockades synergistically prolonged the survival of tumor-bearing mice [98]. Finally, targeting the inhibitory receptor NKG2A may be of interest, since this molecule is expressed on tumor-infiltrating NKG2A^+^ CD8^+^ T_RM_ cells and the blockade of NKG2A potentiates CD8^+^ T cell immunity when combined with cancer vaccines and PD-1 blockade [99,100].

### 3.4. Costimulatory Receptors

Besides inhibitory receptors, CD8^+^ T_RM_ cells can express various costimulatory molecules on their cell surface, which play a key role in regulating immune responses. Most co-signaling molecules are members of the immunoglobulin superfamily (e.g., CD28 and ICOS) and the tumor necrosis factor receptor (TNFR) superfamily [101]. The costimulatory members of the TNFR superfamily, including 4-1BB (CD137), OX40 (CD134), CD27, and glucocorticoid-induced TNFR-related protein (GITR, CD357), are of interest in this respect due to their targetability using agonistic antibodies.

4-1BB regulates effector CD8^+^ T cell accumulation in the lungs through a TRAF1-, mTOR-, and antigen-dependent mechanism to enhance T_RM_ cell formation during respiratory influenza virus infection [102]. Combining 4-1BB agonism with the PD-L1 blockade could increase tumor-infiltrating CD103^+^CD8^+^ T_RM_ cells, thereby enhancing tumor regression [103]. Moreover, targeting OX40 promotes lung-resident memory CD8^+^ T cell populations that protect against respiratory poxvirus infection [104]. The targeting of GITR by an agonistic antibody in patients with advanced cancer and in mice with advanced tumors demonstrated that GITR agonism promotes effector T cell functions and hampers suppression by circulating and intra-tumoral regulatory T cells [105]. GITR signaling can also advance T_RM_ cell formation during respiratory infection with influenza virus [106,107].

Costimulatory molecules expressed by most CD8^+^ T_RM_ cell subsets are CD27 and CD28, and in non-small cell carcinoma, the CD69^+^CD8^+^ TILs express CD27 and CD28 [108]. Agonistic CD28 stimulation ex vivo boosted TNF production by CD69^+^CD8^+^ TILs, while the addition of CD27 costimulation further enhanced TNF and/or IFN-γ production. These data suggest that agonistic stimulation of costimulatory receptors may improve the therapeutic efficacy of cancer immunotherapy by targeting T_RM_ cell populations and by enhancing their expansion and/or cytokine production.

### 3.5. Chemokines and Chemokine Receptors

Chemokines control the migration of a wide variety of cell types, including CD8^+^ T cells. CD8^+^ T_RM_ cells can express many different chemokine receptors, such as CXCR3, CXCR5, and CXCR6. CXCR3 mediates the migratory capacities of T cells into tissues during T_RM_ cell development but is also directly involved in the differentiation of CD8^+^ T cells in response to antigen [109,110]. In addition, regulatory T cells are specifically recruited to local inflammatory sites via the chemokine receptor CXCR3 and promote the generation of CD8^+^ T_RM_ cells via TGF-β in the microenvironment [111]. During chronic lymphocytic choriomeningitis virus (LCMV) infection, virus-specific resident CD8^+^ T cells in lymphoid tissues acquire a PD-1^+^TCF1^+^CXCR5^+^Tim-3^−^ stem-like phenotype [112]. These cells proliferate and give rise to the more terminally differentiated PD-1^+^CXCR5^-^Tim-3^+^ CD8^+^ T cells. Both subsets were found to have limited recirculation capacity as shown by parabiosis experiments. In the lungs, CXCR6 regulates the localization of CD8^+^ T_RM_ cells to the airways [113]. Interstitial-resident memory CD8^+^ T cells sustain frontline epithelial memory in the lungs. This process is driven by CXCR6 that is expressed uniquely on T_RM_ cells, but not T_EM_ cells [37]. In the liver, CXCR6 is essential for T_RM_ cell development and maintenance by binding CXCL16 secreted by liver endothelial cells. Glycolipid-peptide vaccination induces liver-resident memory CD8^+^ T cells that protect against rodent malaria [114]. Vaccination-induced intrahepatic malaria-specific CD8^+^ T_RM_ cells express CXCR6 and CD101, and these cells can be numerically increased via vaccine boosting. Topical application of chemokines can be used as a therapeutic strategy to direct tissue-specific T_RM_ cell formation using a prime-pull approach. In this approach, a systemic T cell response is first elicited by parenteral vaccination, followed by recruitment of activated virus-specific T cells into the tissue by a single topical application of chemokines [115]. Here, the ligands of CXCR3, CXCL9 and CXCL10, were topically applied and mediated T cell recruitment and T_RM_ cell formation in the skin and female reproductive tract (FRT) [6,115]. Thus, protective T_RM_ cell population formation can be induced locally by targeted application of chemokines to direct T cell migration and differentiation. The development of agonistic and antagonistic small molecules targeting chemokine receptors may provide additional means to selectively regulate T cell migration and ultimately T_RM_ cell lodgment [116,117].

### 3.6. Cytokines and Cytokine Receptors

Cytokines play an important role in the differentiation and maintenance of CD8^+^ T_RM_ cells. TGF-β induces the expression of the integrin CD103 on T_RM_ cells, which allows for retention of T_RM_ cells in epithelial tissues, possibly through the interaction with E-cadherin on epithelial cells [6,118]. T_RM_ cells in various mucosal tissues require TGF-β for their maintenance [6,118,119], and competition for active TGF-β allows for selective retention of antigen-specific T_RM_ cells in the epidermal niche [120]. Comparative analysis revealed a role for TGF-β in shaping the residency-related transcriptional signature in CD8^+^ T_RM_ cells [121]. In this context, TGF-β signaling in T_RM_ cells can be further modulated via external signals. Sensing of ATP via P2RX7 promotes CD8^+^ T_RM_ cell generation by enhancing their sensitivity to TGF-β [122]. Moreover, monocyte-produced IL-10 induces the release of surface-bound TGF-β, which in turn induces CD103 upregulation on T cells. IL-10-mediated TGF-β signaling may therefore have a critical role in the generation of T_RM_ cells following vaccination [123]. TGF-β is dispensable in the formation of T_RM_ cells at non-mucosal sites. Consequently, targeting of TGF-β signaling could constitute an attractive approach to specifically modulate T_RM_ cell formation and maintenance at mucosal sites. In line with this, a blockade of TGF-β decreases the induction of T_RM_ cells after mucosal vaccine immunization [124].

Similar to their circulating counterparts, T_RM_ cells in the skin, lungs, liver, salivary glands, and kidneys require IL-15 for their maintenance [12,125,126]. In the skin, IL-15 and IL-7 produced by hair follicles maintain T_RM_ cells in the vicinity of these structures [127], which corresponds to the expression of CD122 (IL-15/2Rβ) and CD127 (IL7Rα) on these T_RM_ cells. Moreover, IL-15 complexes induce the migration of resting memory CD8^+^ T cells into mucosal tissues and enhance the establishment of T_RM_ cells within these tissues [128]. However, IL-15 may not be essential for T_RM_ cells in all tissues, as pathogen-specific T_RM_ cell populations in the pancreas, female reproductive tract, and small intestine are maintained independently of IL-15 [125]. Thus, given its role in both lodgment and maintenance of T_RM_ cells, targeting IL-15 may improve the establishment of protective resident populations at certain tissue sites.

### 3.7. Transcription Factors

The differentiation of naïve CD8^+^ T cells into different memory CD8^+^ T cell subsets constitutes a tightly regulated process under the control of multiple transcription factors. These transcription factors in part integrate extracellular signals and thereby provide insight into the signals that promote T cell residency in non-lymphoid sites. The transcription factors Hobit and Blimp-1 instruct a tissue-residency program that allows for the long-term retention and maintenance of T_RM_ cells within peripheral tissues [17]. The expression of the transcription factor Hobit can be induced by IL-15 signaling in a T-bet-dependent manner, which may thereby shape the transcriptional profile of T_RM_ cells. Other transcription factors implicated in T_RM_ cell development and maintenance include Runx3, Notch, Bhlhe40, AhR, and Nur77 [11,15,16,129,130]. Transcriptional profiling has revealed transcription factors that identify functionally distinct T_RM_ cell subsets. Blimp-1 and Id3 expression delineate distinct tissue-resident T cell subsets in the small intestine, where Id3^hi^ Blimp-1^lo^ T_RM_ cells exhibit heightened multifunctionality and memory potential [34]. While transcription factors are notoriously difficult to target for therapeutic purposes, preclinical studies have assessed their role in T_RM_ cell formation and functionality using transgenic systems. In a study by Milner et al., the overexpression of Runx3 in adoptively transferred CD8^+^ T cells inhibited tumor outgrowth and prolonged survival of mice in a preclinical model of melanoma [11]. In contrast, CD8^+^ T cells lacking Runx3 expression did not accumulate in the tumor microenvironment, resulting in uncontrolled tumor growth and reduced survival. Thus, targeting transcription factors to promote tissue residency could be used to enhance vaccine efficacy or adoptive cell therapy treatments that target cancer.

Certain transcription factors regulating T_RM_ cell development directly integrate extracellular signals. For instance, the activity of AhR, a crucial regulator in maintaining intraepithelial lymphocyte numbers in both the skin and the intestine, can be induced by dietary components, such as those present in cruciferous vegetables, providing a mechanistic link between dietary compounds and the intestinal immune system [131]. Moreover, the transcription factor Notch, which controls the maintenance of T_RM_ cells in the lungs [16], is directly induced by corresponding ligands on neighboring cells. In contrast, the expression of the transcription factor Bhlhe40, which is important for the mitochondrial fitness of T_RM_ cells (as will be discussed hereafter) can be inhibited by local PD-1 signaling [15]. Consequently, the activity of multiple transcription factors important for T_RM_ cell development and function can be modulated directly via extracellular signals. Local and temporal provision of these signals may aid future immunization strategies aiming to induce protective T_RM_ cell populations. Furthermore, the advent of transgenic T cell therapies may provide a platform to equip adoptively transferred T cells with a transcriptional program favoring tissue residency and persistence.

### 3.8. Metabolic Reprogramming

T_RM_ cells undergo tissue-specific metabolic adaptations to persist at tissue sites with restricted nutrient availability. Given their localization within the tissue, T_RM_ cells differ in their access to nutrients and oxygen in comparison to their circulating counterparts. T_RM_ cells in the liver specifically upregulate the hypoxia-inducible transcription factor HIF-2α [132], suggesting that T_RM_ cells primarily reside within hypoxic regions of the liver. Effector responses and survival of liver T_RM_ cells further depend on HIF-2α, indicating a direct link between metabolic conditions, functionality and maintenance of T_RM_ cells. In the skin, T_RM_ cells utilize mitochondrial β-oxidation of exogenous FFA taken up via fatty acid binding protein (FABP) 4 and 5 for their survival and functionality [26]. T_RM_ cells at other tissue sites may also rely on fatty acid oxidation but express different FABP isoforms, with FABP1 supporting the establishment of T_RM_ cells in the liver [27]. Similarly, T_RM_ cells in gastric adenocarcinoma rely on fatty acid oxidation for their survival and may compete with tumor cells for the uptake of fatty acids [133]. Circulating CD8^+^ T_CM_ cells also depend in part on mitochondrial fatty acid oxidation for cellular metabolism. However, rather than acquiring exogenous fatty acids like T_RM_ cells, T_CM_ cells appear to synthesize fatty acids from exogenous glucose and glycerol to support fatty acid oxidation and cellular longevity [134,135,136]. Consequently, targeting fatty acid uptake and oxidation might represent an attractive approach to specifically modulate T_RM_ cell longevity and functionality. The mitochondrial metabolism of T_RM_ cells is, in part, regulated by the transcription factor Bhlhe40, which supports the TCA cycle activity and oxidative phosphorylation in T_RM_ cells [15]. The efficacy of the PD-L1 blockade in reinvigorating tumor-infiltrating T_RM_ cells depends on Bhlhe40, and Bhlhe40 might therefore constitute a nexus between immunomodulatory signals, metabolism, and functionality of tissue-resident T cell populations. Interestingly, Lin et al. found that the PD-L1 blockade promoted FABP4/5 expression, fatty acid uptake, and survival of tumor-infiltrating T_RM_ cells in gastric cancer [133]. The induction of metabolic changes in tissue-resident T cell populations may therefore contribute to the effectiveness of an immune checkpoint blockade. Given that T_RM_ cells in many tissues express high amounts of inhibitory receptors (including PD-1, Tim-3, and CTLA-4) [6,16,43,137], an immune checkpoint blockade may allow for the manipulation of both functionality and metabolic fitness of T_RM_ cells. Direct targeting of metabolic pathways sustaining T_RM_ cells could be another way to regulate T_RM_ cell maintenance. The pharmacological inhibition of mitochondrial β-oxidation in vivo decreases the maintenance and survival of T_RM_ cells in the skin [26]. The external modulation of T_RM_ cell metabolism could thus be employed to either increase or decrease local T_RM_ cell populations. As T_RM_ cells have been implicated in the pathogenesis of various inflammatory diseases such as psoriasis in the skin [138], targeting the specific metabolic requirements of these cells might thus also constitute a potential therapeutic approach to treat auto-inflammatory disorders.

## 4. Conclusions

The tissue environment, cellular interaction partners, local cytokines and chemokines, as well as antigen triggering play roles in the development of CD8^+^ T_RM_ cells and shape the phenotype of these cells. Consequently, CD8^+^ T_RM_ cells arising at different sites and responding to different infections are characterized by diverse phenotypes. This phenotypic heterogeneity is exemplified by the expression of various cell surface molecules, transcription factors, and diverse metabolic and signaling pathways. The heterogeneous nature of CD8^+^ T_RM_ cells may allow for the specific targeting of T_RM_ cell subsets in selected organs for immunotherapy. Thus far, memory CD8^+^ T cell subsets are mostly nonspecifically targeted, while targeting of specific CD8^+^ T_RM_ cell subsets may be more beneficial. However, this remains challenging. A better understanding of the molecular determinants of CD8^+^ T_RM_ cell heterogeneity is therefore crucial to improve the design of vaccines and immunotherapies aiming to harness the protective capacity of T_RM_ cell subsets.

## Figures and Tables

**Figure 1 cells-10-00164-f001:**
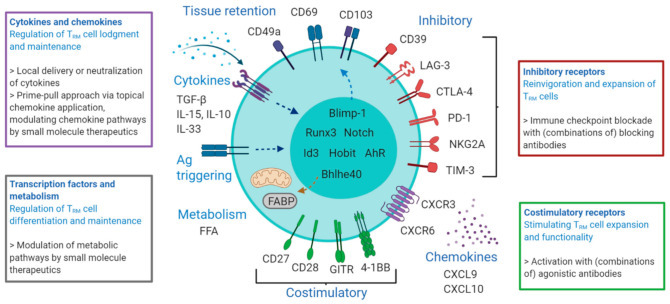
**Heterogeneity and therapeutic targeting of CD8^+^ T_RM_ cells.** Integrins, such as CD69, CD49a, and CD103, mediate the tissue retention of CD8^+^ T cells. Multiple factors, including the cytokine environment, and antigen-triggering and cellular interactions differentially regulate the formation of heterogeneous T_RM_ cell subsets that differ by their costimulatory and inhibitory cell surface molecules, cytokine and chemokine receptors, transcription factors, and intracellular metabolic and signaling pathways. The unique characteristics of the heterogeneous phenotypes and functions of CD8^+^ T_RM_ cells can be exploited for therapeutic targeting to improve viral immunity and immunotherapies in malignant diseases. Memory CD8^+^ T cell subsets expressing the inhibitory molecules PD-1, CD39, LAG-3, CTLA-4, NKG2A, and Tim-3 can be successfully targeted by antibodies to unleash effector functions, but specific targeting of CD8^+^ T_RM_ cells remains challenging. In addition, CD8^+^ T_RM_ cells can express a variety of costimulatory molecules including CD27, CD28, GITR, and 4-1BB. Agonistic antibodies to these molecules can improve the expansion and cytokine production of CD8^+^ T_RM_ cells. T_RM_ cell lodgment and maintenance are possible via targeting the cytokine and chemokine receptors on CD8^+^ T_RM_ cells by (local) delivery of the cytokines and chemokines (e.g., topical application) or by modulating chemokine pathways using small molecule therapeutics. Additionally, tissue residency could be promoted by targeting transcription factors such as Runx3 in adoptive cell therapy treatments that target cancer. Moreover, CD8^+^ T_RM_ cells have specific metabolic requirements and targeting these might constitute a potential therapeutic approach. Figure created with BioRender.

## Data Availability

Not applicable.

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
