# Peer review of "Functional Heterogeneity and Therapeutic Targeting of Tissue-Resident Memory T Cells"

_cells, 2021, doi:10.3390/cells10010164_

Round 1

Reviewer 1 Report

Comments:

I would like to thank the authors for submitting this review manuscript on Tissue-resident memory T cells. However, there are a few manuscripts already published which focus on this subject (1, 2). It is therefore crucial to highlight how this manuscript differs from the published ones.

  • One suggestion might be to focus on the recent papers using single cell sequencing that have dissected the heterogeneity of T cells in tissues (3, 4).
  • The authors could also discuss in detail about what questions need to be addressed in a separate section. There are some limitations in studying Trm in the context of humans. The authors could highlight that and suggest ideas to address them.
  • The authors might also highlight their findings about functional heterogeneity in various tissues in a diagram.
  • Another figure might represent the therapeutic targets hypothesized and how it would benefit in case of infection and cancer.

References:

  1. Masopust D, Soerens AG. Tissue-Resident T Cells and Other Resident Leukocytes. Annu Rev Immunol. 2019 Apr 26;37:521-46.
  2. Szabo PA, Miron M, Farber DL. Location, location, location: Tissue resident memory T cells in mice and humans. Sci Immunol. 2019 Apr 5;4(34).
  3. Kurd NS, He Z, Louis TL, Milner JJ, Omilusik KD, Jin W, et al. Early precursors and molecular determinants of tissue-resident memory CD8(+) T lymphocytes revealed by single-cell RNA sequencing. Sci Immunol. 2020 May 15;5(47).
  4. Tkachev V, Kaminski J, Potter EL, Furlan SN, Yu A, Hunt DJ, et al. Spatiotemporal single-cell profiling of gastrointestinal GVHD reveals invasive and resident memory T cell states. bioRxiv. 2020:2020.07.20.212399.

Author Response

We would like to thank the reviewer for the kind words and the suggestions to improve our manuscript. The primary focus of our review is the therapeutic targeting of TRM cells. Tor our knowledge, this particular topic has not been reviewed extensively in the literature thus far, and we believe our manuscript therefore presents a novel and timely overview of the current knowledge in this field. In order to emphasize this focus of our manuscript, we have adapted the abstract and the introduction of our manuscript. Additionally, as suggested by the reviewer, we have amended the section on TRM heterogeneity with the recent findings gained by scRNAseq (lines 137-141). Furthermore, we have acknowledged the challenges in investigating TRM cells in human tissues (lines 91-93). Lastly, we have included a more extensive figure to display the potential therapeutic targets on TRM cells in greater detail.

Reviewer 2 Report

The review report "Functional heterogeneity and therapeutic targeting of tissue-resident memory T cells", written by Esmé et al., covers a variety of topics of TRM cells, from their differentiation to their prospect for therapeutic purposes, including plenty of current discoveries. This manuscript is basically well written, and intelligible to readers in other disciplines. However, I felt that addition of one or two more figures (e.g., a predicted model for TRM cell development from progenitors, roles of transcription factors etc.) will certainly supports the readers' understanding of TRM cell biology.

Author Response

We would like to thank the reviewer for the kind words regarding our manuscript. As suggested also by reviewer 1, we have included a more extensive figure to display the potential therapeutic targets on TRM cells in greater detail.

Round 2

Reviewer 1 Report

I would like to thank the authors for incorporating the changes.